# Seasonal Abundance and Diversity of *Culicoides* Biting Midges in Livestock Sheds in Kanchanaburi Province, Thailand

**DOI:** 10.3390/insects15090701

**Published:** 2024-09-14

**Authors:** Arunrat Thepparat, Naoto Kamata, Padet Siriyasatien, Waranya Prempree, Kannika Dasuntad, Boonruam Chittsamart, Sunisa Sanguansub

**Affiliations:** 1Department of Agricultural Technology, Faculty of Science, Ramkhamhaeng University, Huamark, Bangkapi, Bangkok 10240, Thailand; arunratthepparat@gmail.com (A.T.); kannikadasuntad@gmail.com (K.D.); 2The University of Tokyo Chiba Forest, Graduate School of Agricultural and Life Sciences, The University of Tokyo, Kamogawa 299-5503, Japan; kamatan@uf.a.u-tokyo.ac.jp; 3Center of Excellence in Vector Biology and Vector Borne Diseases, Department of Parasitology, Faculty of Medicine, Chulalongkorn University, Bangkok 10330, Thailand; padet.s@chula.ac.th; 4Department of Biology, Faculty of Science, Ramkhamhaeng University, Huamark, Bangkapi, Bangkok 10240, Thailand; min5834@gmail.com; 5Faculty of Public Health, Bangkok Thonburi University, Bangkok 10700, Thailand; boonruamc@gmail.com; 6Department of Entomology, Faculty of Agriculture at Kamphaeng Saen, Kasetsart University, Kamphaeng Saen Campus, Nakhon Pathom 73140, Thailand

**Keywords:** body size, *Culicoides* spp., dispersal ability, livestock, neighboring effect, poultry, seasonal abundance, weather factor

## Abstract

**Simple Summary:**

The emergence of midge-borne diseases such as lumpy skin disease in Thailand highlights the necessity of understanding the seasonal abundance and spatial distribution of *Culicoides* biting midges. Population dynamics, species composition, and host preferences of *Culicoides* species were studied in Kanchanaburi, Thailand. Nine UV light traps were deployed to collect midges in nine animal sheds over the course of a year, revealing abundant species such as *C. oxystoma* Kieffer and *C. peregrinus* Kieffer, *C. arakawae* Arakawa, and *C. imicola* Kieffer. These species demonstrated preferences for different farm animals and peaked in trap captures from the late rainy season to early winter. Trap catches were positively influenced by relative humidity and temperature, while they exhibited a negative impact from wind speed. This study provides important data for disease management and prevention, emphasizing the impact of these vectors on both animal and human health. Overall, this research underscores the importance of ongoing studies into *Culicoides* behavior, ecology, and disease transmission potential to inform effective control strategies.

**Abstract:**

*Culicoides* biting midges were collected using UV light traps from nine livestock farms in Kanchanaburi Province, Thailand. Collections were made one night per month from June 2020 to May 2021 to determine the seasonal changes and spatial distribution of the *Culicoides* assemblage. The influence of four environmental factors (temperature, rainfall, humidity, and wind speed), types of animals in each shed (cattle, pigs, and avians), and neighboring animals (those in the other sheds and their proximity) were assessed. A subsample of 130,670 out of a total of 224,153 specimens collected was identified and counted. The collections were predominantly female (76.9%), though males were also commonly collected (23.1%). The dominant species included *C. peregrinus* (97,098 individuals; 43.0%), *C. oxystoma* (55,579; 24.6%), *C. arakawae* (45,996; 20.4%), and *C. imicola* (15,703; 7.0%), while other species accounted for the remaining 9777 individuals (4.4%). Trap captures were strongly influenced by seasons and animal sheds. Cattle exhibited the greatest effect on the community, likely due to their large biomass. Humidity and temperature had a positive effect on trap captures, whereas wind speed exhibited a negative effect. Cattle positively influenced all major species, except for *C. arakawae*, which showed a positive association with avians. A “neighboring effect” was also observed. Additionally, the lowest dispersal ability of *C. arakawae* was suggested.

## 1. Introduction

*Culicoides* Latreille species (Diptera: Ceratopogonidae) are regarded as primary vectors of numerous medico-veterinary diseases. These small, blood-sucking biting midges have a body size ranging from 1 to 3 mm [1]. They have a global distribution with approximately 1450 described species to date [2]. These insects are known to cause allergies, skin rashes, and biting nuisances to humans and animals. Moreover, they serve as vectors for numerous disease-causing viruses including African horse sickness (AHS), bluetongue virus, Schmallenberg virus, epizootic hemorrhagic disease virus, West Nile virus, Akabane virus, and Oropouche virus as well as protozoan and nematode parasites [3]. These diseases have significant impacts on both human and animal health, leading to substantial economic losses in the livestock industry and posing risks to public health through zoonotic transmission.

In Thailand, 103 species of *Culicoides* biting midges have been reported [4,5,6], including several known vectors of livestock diseases such as *C. oxystoma*, *C. peregrinus*, and *C. orientalis* [7]. More recently, studies have identified *Culicoides* species in Thailand that are potential vectors of human pathogens, including Leishmania [8,9]. The emergence of midge-borne diseases such as AHS [10] and lumpy skin disease (LSD) [11] in Thailand underscores the need for information on the species involved, including their host preferences, relative abundance, and seasonality.

Despite their significance as disease vectors to humans and the animal economy, little is known about the ecology of *Culicoides* species in Thailand. Studying species abundance in animal sheds is crucial to understanding how livestock are affected by vectors, pests, and the pathogens they transmit. Disease management authorities need this information if they are to take appropriate action to manage these threats. Local communities, especially farmers residing near or under livestock sheds that can serve as reservoirs for livestock diseases, are at risk of the emergence of zoonotic diseases. The study areas are surrounded by livestock sheds and irrigated regions suitable for livestock rearing. Although insecticides are applied twice a month in animal sheds, the biting midge population and other insects remain unaffected.

Population abundance, seasonal dynamics, and host preferences of vector species are critical factors in the epidemiology of vector-borne diseases. However, few studies are available on the relative abundance, seasonality, or host preferences of *Culicoides* species in Southeast Asia. The recent emergence of vector-borne diseases in this region emphasizes the need to gather this data to aid in the development of effective disease management strategies.

The objective of this research was to determine the seasonal changes in the assemblage of *Culicoides* biting midges in the animal sheds in Kanchanaburi Province, Thailand. Specifically, this study aimed to assess the effects of environmental factors, such as temperature, rainfall, relative humidity, and wind speed, as well as the types of animals present in each shed, on the species composition of *Culicoides* biting midges. The impact of neighboring sheds was also evaluated. To achieve this, one-night UV light trapping was conducted over the course of a year, from June 2020 to May 2021. The captured specimens were subsequently subjected to morphological identification, and environmental data were recorded on the same nights as the trapping. The effects of environmental factors, animal types, and neighboring sheds were analyzed. Additionally, the dispersal ability of *Culicoides* biting midges was discussed based on the results. The findings from this research will provide valuable information for the development of targeted vector control strategies aimed at minimizing the impact of midge-borne diseases affecting animals in the region.

## 2. Materials and Methods

### 2.1. Biting Midge Collection

This study was conducted in animal sheds in Tha Maka district, Kanchanaburi province, Thailand (13°52′ N; 99°46′ E), which is located in the western part of Thailand bordering Myanmar. Midges were collected using UV light traps set near animal sheds on 9 farms within a radius of 200 m (Figure 1). Traps were positioned consistently relative to the host animals and were operated for one night every month from early evening to dawn (6 p.m. to 6 a.m.) [12] between June 2020 and May 2021, resulting in a total of 108 collections (9 traps × 12 months). Temperature, relative humidity, rainfall, and wind speed were continuously measured by using a digital hygrometer (Kestrel 3000, Richard Paul Russell Ltd., Lymington, UK) from June 2020 to May 2021. Trapping sites and areas where the animals were kept are provided in Figure 1 and details of possible host animals near the trap sites are given in Table 1. The LSD was detected in the cattle housed in the shed near Trap8 in May 2021.

### 2.2. Materials

#### 2.2.1. Sorting of Biting Midges

Insects were collected alive in a mesh bag and subsequently transferred into 70% ethanol in a 100 mL plastic bottle when the trap was cleared. Collections were transported to the laboratory, where insects other than *Culicoides* biting midges were removed from the bottles primarily based on their characteristic wing patterns using a stereomicroscope (Olympus, Tokyo, Japan) (Appendix A Appendix A). Additional morphological features, such as the spermatheca, thorax, and legs, were also examined to confirm identification [13]. The number of individuals of the *Culicoides* biting midges in each collection was recorded. Individual *Culicoides* biting midges were taken from the jar and identified until the number examined exceeded approximately 2000. For collections in which not all individuals were examined, the data were extrapolated to estimate the overall composition of the collection, assuming that the proportions of each species were equivalent to those in the subsample.

#### 2.2.2. Morphological Identification of *Culicoides* Species

The *Culicoides* biting midges collected were identified according to species using a stereomicroscope. Most specimens were identified in 70% ethanol based on the descriptions and illustrations provided by Wirth and Hubert [14] and Dyce et al. [15]. Representative specimens were mounted onto glass slides as described by Bellis et al. [16] using Hoyer’s medium. Mounted specimens were identified using the key and descriptions from Wirth and Hubert [14].

### 2.3. Statistical Analysis

All the statistical analyses were conducted using R version 4.1.2 [17].

The number of individuals for each of the three types of animals were used for the statistical analyses: cattle (including cows and dairy cattle), pigs, and avians (including duck, chicken, game fowl, and other birds).

To evaluate the effect of neighboring animal sheds and their proximity on the *Culicoides* assemblage (hereafter referred to as the “neighboring effect”), correlation between geographical distance and dissimilarity of *Culicoides* assemblage between traps was determined using the Mantel test. Geographical distance was calculated using a library ‘geosphere’ (ver. 1.5-19) [18] with the coordinates of the two traps (Appendix A Appendix A). The Bray–Curtis dissimilarity index and differences in abundance of four major species—*C. peregrinus*, *C. oxystoma*, *C. arakawae*, and *C. imicola*— between two traps were utilized for this analysis.

Permutational multivariate analysis of variance (PERMANOVA) was employed to determine the effects of date and traps on the community structure of the *Culicoides* assemblage along with tests of homogeneity of dispersions (PERMDISP). Another PERMANOVA was conducted to assess the effects of dates (DSIN, DCOS), environmental factors (temperature, humidity, rainfall, and wind speed), and the three types of animals (cattle, pigs, and avians) on the community structure of the *Culicoides* assemblage.

A library ‘vegan’ (ver. 2.7) [19] was used to obtain the Bray–Curtis dissimilarity index and to conduct the PERMDISP and PERMANOVA.

A generalized linear mixed model (GLMM) was employed to determine the effects of the four environmental factors and the three types of animals on the numbers of *Culicoides* collected. To determine the effects of environmental factors, traps were used as a random effect. In contrast, date was employed as a random effect to evaluate the effect of the three types of animals. The response variable included total trap captures or abundance of each of the four major *Culicoides* species collected. A library ‘lme4’ (ver. 1.1-35.1) was used for the GLMMs [20].

## 3. Results

### 3.1. Species Composition of Culicoides Biting Midges

A total of 130,670 specimens of *Culicoides* were identified from an overall collection of 224,153 collected; however, 2040 individuals could not be identified due to a lack of keys (damaged) or descriptions (unknown) (Table 2). Among the identified specimens, 76.9% were female and 23.1% were male. A total of 31 species were identified, with the most abundant species being *C. peregrinus* (43.0%), followed by *C. oxystoma* (24.6%), *C. arakawae* (20.4%), and *C. imicola* (7.0%). The catch data of the other species, including unidentified individuals, totaled 9777 (4.4%).

There was considerable variation among traps in the numbers of captures (Figure 2). The traps located on the farms with cattle captured a greater number of midges than those in the other locations. *Culicoides peregrinus* was the most abundant species in the collections from farms with cattle. Trap captures of *C. arakawae* tended to be greater in traps located on the farms with avians, with the exception of Trap9. Trap3, which was the only trap set on the farm with pigs, was predominated by *C. oxystoma* (65.7%). *Culicoides imicola* was also abundant on farms with cattle.

### 3.2. Factors Influencing Culicoides Biting Midge Assemblage

Great seasonal variations were also observed in trap captures (Figure 3). Captures peaked in November, decreased sharply in December, remained low through February, and started to increase again in March. Monthly variation was greater than the variation among traps within each month.

Figure 4 illustrates the relationship between the geographical distance between traps and community dissimilarity. The Bray–Curtis dissimilarity index value, evaluated using the four major species, tended to increase with the geographical distance, indicating that the distance between the traps influenced the community composition (*p* < 0.05) (Figure 4a). Differences in the abundance of *C. imicola*, *C. peregrinus*, and *C. oxystoma* also tended to increase with geographical distance, although the correlations were marginally significant (*C. imicola*, *p* = 0.077; *C. peregrinus*, *p* = 0.053; *C. oxystoma*, *p* = 0.092) (Figure 4b,c,e). However, no relationship was observed with *C. arakawae* (*p* = 0.542) (Figure 4d).

The variance of dispersion was not significantly different across both dates and traps (dates, *p* = 0.690; traps, *p* = 0.142; PERMDISP) although the effects of dates and traps were significant (dates, *p* < 0.001; traps, *p* < 0.001; PERMANOVA) (Appendix A Appendix A), indicating differences in the community composition among traps and dates. The results of another PERMANOVA assessing the effects of animals and environmental factors indicated that the effect of cattle was the greatest among the six significant explanatory variables, followed by humidity and rainfall (*p <* 0.05, PERMANOVA) (Table 3). The effect of wind speed was marginally significant (*p* = 0.073), while the effects of temperature and pigs were not significant.

Table 4 illustrates results of the GLMM assessing the effects of environmental factors on the numbers of midges collected. Temperature and humidity had positive effects on trap captures, while wind speed had a negative effect. The effect of rainfall differed between midge species, with *C. imicola* and *C. oxystoma* experiencing negative effects, while *C. peregrinus* and *C. arakawae* saw positive effects. Humidity had the greatest effect among the four environmental factors for total midges as well as for *C*. *imicola* and *C. oxystoma*. In contrast, the effect of rainfall was positive and the greatest for *C. peregrinus*, while the effect of wind speed was negative and the most pronounced for *C. arakawae*.

Table 5 illustrates results of the GLMM assessing the effects of animals on the numbers of midges collected. All three animals had significant effects on all four major midge species, including total midges, with the exception of the effect of cattle on *C. arakawae*. Cattle had positive effects on the trap captures for all species, including total midges, except for *C. arakawae*, where the effect was not significant. For *C. arakawae*, only the effect of avians was positive (*p* < 0.001).

## 4. Discussion

Population abundance, seasonal dynamics, and host preferences of vector species are critical factors in the epidemiology of vector-borne diseases. This year-round study indicates that the assemblage of *Culicoides* biting midges was influenced by both nearby host animals and environmental factors (Table 3). In addition to these two major factors, the presence of host animals in neighboring animal sheds and their proximity also influenced the midge assemblage (Figure 4). This finding suggests that *Culicoides* biting midges likely dispersed among the animal sheds due to attraction to the animals in the adjacent sheds.

Trap captures of *Culicoides* biting midges exhibited great seasonal variation (Figure 4), as well as variation among traps (Figure 3). Specifically, populations increased towards the end of the dry season (March), peaked at the end of the rainy season and the beginning of winter (October–November), experienced a sharp decline in December, and remained low until February when temperatures were lower. These results suggest that the seasonal variation in trap captures likely depended on fluctuations in environmental factors.

It has been reported that temperature, wind speed, light intensity, lunar cycles, relative humidity, changes in barometric pressure, and other weather conditions can influence the flight activity of biting midges [21]. In this study, the effects of temperature, rainfall, relative humidity, and wind speed on light trap collections were investigated. The results of the GLMM, with traps as a random effect, confirmed the significant effects of all four factors on the trap captures (Table 4). Temperature and humidity were positively correlated with the numbers of midges, which aligns with the previous report [22].

Conversely, the effect of wind speed was negative, which is consistent with another report [23]. This negative effect is likely due to the fact that *Culicoides* biting midges are not strong flyers and cannot fly at wind speeds exceeding 1.6 m/s [24] or 2.5 m/s [25]. Even though the wind speed recorded in this study was lower than 0.7 m/s, the wind speed still had a negative effect on the number of trap captures of *Culicoides* biting midges [25]. Generally, wind tends to be weaker at lower latitudes where the temperature difference across latitudes is smaller. *Culicoides* biting midges may be less adapted to strong winds and more sensitive to wind speed compared to those in higher latitudes.

Among the four major species in our study, *C. arakawae* exhibited the strongest negative impact of wind (Table 4). The “neighboring effect” of *C. arakawae* was the weakest among the four major species, as the dissimilarity between the traps did not show a positive relationship with the geographical distance (Figure 4). These results suggest a poor dispersal ability of *C. arakawae*. In fact, *C. arakawae* has the smallest body size and wing size among the four major species (Appendix A), which likely limits its flight endurance.

Larvae of *Culicoides* species inhabit semi-aquatic environments such as freshwater marshes and swamps, shallow margins of ponds, streams, rivers, bogs and peat lands, as well as tree holes and other natural cavities [26]. Therefore, it is reasonable to expect that increased rainfall leads to a higher availability of immature habitats, resulting in population increases. However, the effect of rainfall showed the least impact on the total number of traps captures among the environmental factors studied (Table 4). Furthermore, rainfall negatively affected the trap captures of *C. imicola* and *C. oxystoma*. It has been reported that *C. imicola* prefers drier habitats for oviposition [27], which is likely one of the causes for the negative effect of rainfall on *C. imicola*. *Culicoides* species that peaked in months other than October, such as *C. imicola* and *C. oxystoma*, exhibited a negative effect from rainfall possibly because the rainfall data recorded zero precipitation in all months except for October.

Among the four environmental factors, humidity exhibited the greatest effect on trap captures followed by temperature and wind speed. Studies conducted in Jamaica [28] and Florida, USA [29] reported that relative humidity had the least impact on trap captures compared to other environmental variables. In contrast, research in Kenya indicated that optimal conditions for adult biting midge activity occurred at high relative humidity levels [30]. This research was conducted in the seasonal tropics of the Asian monsoon region, where a strong dry season exists. This is likely the direct cause of the pronounced effect of humidity on trap captures among the four environmental factors.

Regarding temperature, there have been instances where temperate-zone *Culicoides* species that appear in the spring and/or fall exhibited negative responses to temperature. However, in this study, the effect of temperature was consistently positive (Table 4), likely because our research was conducted in an environment where temperatures do not decrease to an inhibitory level that would cause diapause, even during the coldest months of December and January. Consequently, higher temperatures are likely to enhance the larval growth of biting midges and adult flight activity among tropical species.

In contrast to seasonal variation, the variation in trap captures among traps was likely influenced by the presence of host animals (Figure 2). Among the three animal types, cattle had the most substantial effects on *Culicoides* biting midges (Table 3 and Table 5); total captures and the four major species, excluding *C. arakawae*, exhibited significantly positive effects from cattle. This observation is likely attributed to their large body size, which supports a considerable population of *Culicoides* biting midges.

Species of *Culicoides* biting midges that are abundant on cattle farms should be considered candidate vectors of cattle diseases, particularly for emerging diseases such as LSD. In this study, *Culicoides peregrinus*, *C. oxystoma*, *C. imicola,* and *C. arakawae* were found to be abundant on farms with cattle (Figure 2). The first three species are known to feed on cattle [21,31] and have been associated with cattle pathogens [3,32], warranting further investigation as potential vectors of LSD and other midge-borne diseases.

Among the four major species, only *C. arakawae* exhibited positive associations with avians (Table 5). In fact, *C. arakawae* is known to attack poultry and is associated with poultry disease [33]. In this study, *C. peregrinus* and *C. oxystoma* were also found to be abundant on farms with avians (Figure 2). However, Jomkumsing et al. [34] demonstrated that *C. arakawae* fed on chickens, while *C. peregrinus*, *C. oxystoma*, and *C. imicola* fed exclusively on mammals (specifically water buffalo and cattle in their study) according to DNA barcoding. Therefore, these major species, other than *C. arakawae*, were likely attracted not to the chickens but to the mammals in the sheds.

*Culicoides oxystoma* was found to be the most abundant on pig farms, suggesting it could be considered a potential vector of pig diseases. However, there is currently no evidence indicating that this species feeds on pigs. In this study, only one trap was set near the shed housing pigs (Table 2, Figure 2). Further research on the *Culicoides* pests associated with pigs is necessary to assess their potential to transmit pig diseases.

This study is limited to a single year. Data for pigs are also limited due to experimental design. To more accurately assess the seasonal abundance of *Culicoides*, it is essential to analyze data over several years to ensure that results are consistent across different years. Establishing a comprehensive database for vector population forecasting and transmission management will be crucial in preventing and minimizing outbreaks of infectious diseases in the future.

## 5. Conclusions

In this study, *C. peregrinus* was identified as the most abundant biting midge species, followed by *C. oxystoma*, *C. arakawae*, and *C. imicola*. The abundance of *Culicoides* biting midges was the highest at the end of the rainy season and the beginning of winter (October–November), and the lowest during winter (December–February). Seasonal changes in trap captures were influenced by environmental factors. The positive effects of humidity and temperature, along with the negative effect of wind speed, were consistently observed, with humidity having the greatest impact. Among the four major species, *C. arakawae* exhibited the strongest negative response to wind speed, which is likely attributed to its small body size, resulting in limited dispersal ability. The presence of cattle positively influenced all four major species, except for *C. arakawae*, possibly due to their large biomass. In contrast, a positive effect of avian hosts was observed only for *C. arakawae*, aligning with the fact that this species exclusively feeds on chickens and is associated with poultry diseases.

## Figures and Tables

**Figure 1 insects-15-00701-f001:**
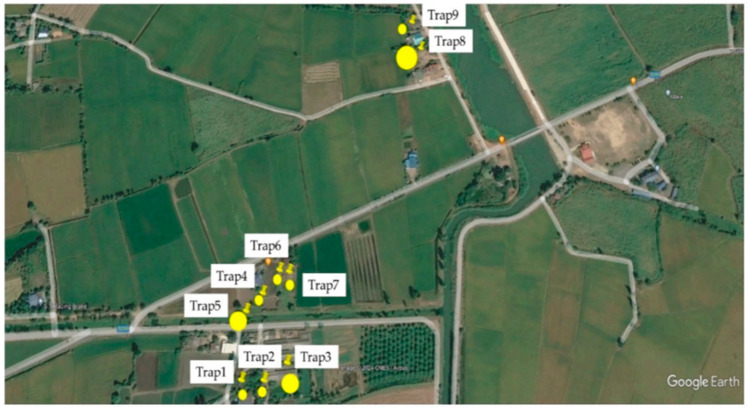
Aerial photo illustrating the positions of UV light traps and the locations of animals within the study area. The endpoints of the pins denote the trap locations, labeled as Trap1–Trap9. The yellow circles indicate the areas where the animals were kept at each site.

**Figure 2 insects-15-00701-f002:**
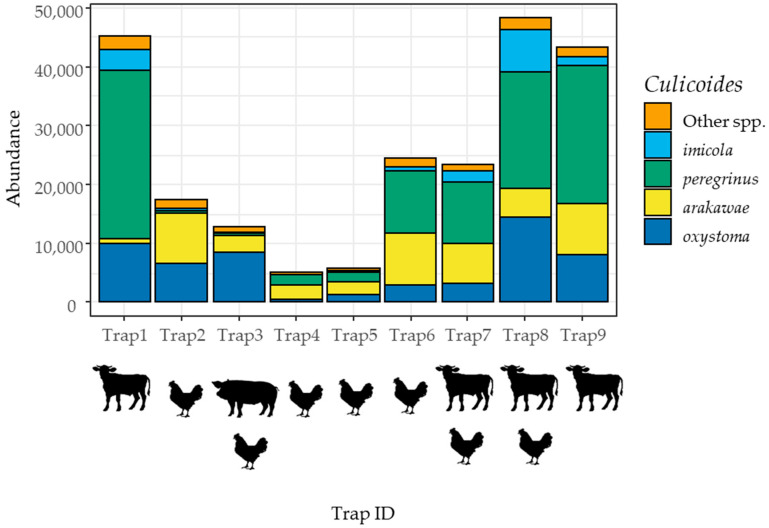
Abundance of four major *Culicoides* biting midge species and other species captured by each of the nine UV light traps in relation to the presence of cattle, pigs, or avians. Total numbers for each trap over the 12 trapping sessions are shown. Trap captures were made using nine UV light traps set once a month from June 2020 to May 2021 in Tha Maka, Kanchanaburi, Thailand.

**Figure 3 insects-15-00701-f003:**
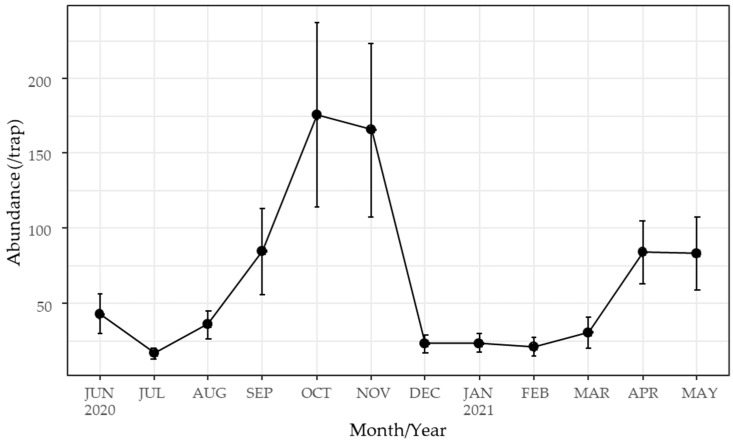
Seasonal changes in the abundance of *Culicoides* biting midges captured by nine UV light traps set once a month from June 2020 to May 2021 in Tha Maka, Kanchanaburi, Thailand. Bar indicates mean ± SE of the nine traps.

**Figure 4 insects-15-00701-f004:**
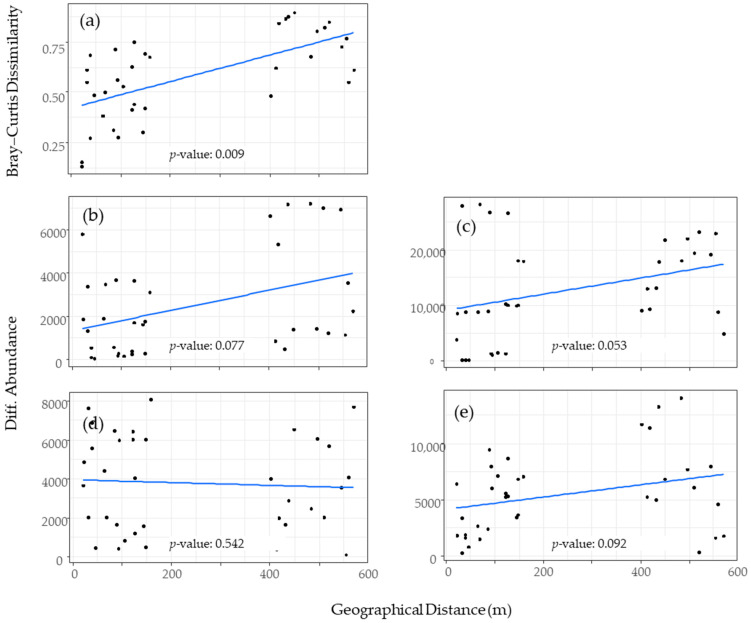
Relationship between geographical distance and dissimilarity indices of *Culicoides* biting midges captured by nine UV light traps. (**a**) Bray–Curtis dissimilarity; (**b**) difference in abundance of *Culicoides imicola*; (**c**) difference in abundance of *Culicoides peregrinus*; (**d**) difference in abundance of *Culicoides arakawae*; (**e**) difference in abundance of *Culicoides oxystoma*. The *p*-values obtained from the Mantel test are shown.

**Table 1 insects-15-00701-t001:** Positions of the nine UV light traps used to capture *Culicoides* biting midges, along with the animals in the vicinity of each trap, and the animal data utilized for statistical analysis (Tha Maka, Kanchanaburi, Thailand; June 2020 to May 2021).

Trap ID	Latitude/ Longitude	Animals	Animal Data Used for Analysis
Trap1	13.8706° N 99.7783° E	3 cows ^1^	Cattle, 3
Trap2	13.8706° N 99.7786° E	100 free-range chickens ^2^ 5 game fowls ^3^	Avians, 105
Trap3	13.8708° N 99.7789° E	100 pigs ^4^ (No pigs in AUG–OCT 2020) 20 free-range chickens ^2^	Pigs, 100 (0, AUG–OCT 2020) Avians, 20
Trap4	13.8714° N 99.7783° E	20 chickens ^2^ 3 game fowls ^3^	Avians, 23
Trap5	13.8717° N 99.7786° E	100 ducks ^5^	Avians, 100
Trap6	13.8719° N 99.7789° E	5 birds ^6^ 20 game fowls ^3^	Avians, 25
Trap7	13.8719° N 99.7786° E	20 chickens ^2^ 5 dairy cattle ^1^ (No cattle in and after MAR 2021)	Avians, 20 Cattle, 5 (0, MAR–MAY 2021)
Trap8	13.8750° N 99.7808° E	3 parrots ^7^ 40 cows	Avians, 3 Cattle, 40
Trap9	13.8750° N 99.7810° E	3 cows	Cattle, 3

^1^ Cow/cattle (Bos taurus), ^2^ Free-range chicken (Gallus gallus), ^3^ Game fowl (Gallus gallus domestica), ^4^ Pig (Sus scrofa domesticus), ^5^ Duck (Anas platyrhynchos), ^6^ Bird (Geopelia striata), ^7^ Parrots (Psittacus torquata).

**Table 2 insects-15-00701-t002:** Number and percentage of identified *Culicoides* biting midge species and extrapolated captures by nine UV light traps set once a month from June 2020 to May 2021 in Tha Maka, Kanchanaburi, Thailand. The species ID for each species is presented.

Scientific Name	Subgenus	ID	Identified Abundance	%	Extrapolated Abundance	%	Rank
*Culicoides actoni* Smith	*Avaritia*	Sp01	210	0.16	320	0.14	12
*Culicoides brevipalpis* Delfinado	*Avaritia*	Sp02	114	0.09	158	0.07	16
*Culicoides brevitarsis* Kieffer	*Avaritia*	Sp03	166	0.13	211	0.09	14
*Culicoides flavipunctatus* Kitaoka	*Avaritia*	Sp04	6	0.00	15	0.01	24
*Culicoides fulvus* Sen and Das Gupta	*Avaritia*	Sp05	259	0.20	386	0.17	11
*Culicoides imicola* Kieffer	*Avaritia*	Sp06	10,721	8.20	15,703	6.95	4
*Culicoides jacobsoni* Macfie	*Avaritia*	Sp07	21	0.02	22	0.01	23
*Culicoides minimus* Wirth and Hubert	*Avaritia*	Sp08	5	<0.001	5	<0.001	27
*Culicoides obscurus* Tokunaga and Murachi	*Avaritia*	Sp09	2	<0.001	2	<0.001	28
*Culicoides orientalis* Macfie	*Avaritia*	Sp10	52	0.04	65	0.03	20
*Culicoides tainanus* Kieffer	*Avaritia*	Sp11	118	0.09	194	0.09	15
*Culicoides wadai* Kitaoka	*Avaritia*	Sp12	1	<0.001	1	<0.001	31
*Culicoides clavipalpis* Mukerji	*Clavipalpis* group	Sp13	109	0.08	111	0.05	17
*Culicoides arenicola* Howarth	*Clavipalpis* group	Sp14	2	<0.001	2	<0.001	29
*Culicoides huffi* Causey	*Clavipalpis* group	Sp15	1548	1.18	1756	0.78	5
*Culicoides innoxius* Sen and Das Gupta	*Hoffmania*	Sp16	743	0.57	993	0.44	7
*Culicoides peregrinus* Kieffer	*Hoffmania*	Sp17	36,950	28.28	97,098	42.99	1
*Culicoides arakawae* Arakawa	*Meijerehelea*	Sp18	30,857	23.61	45,996	20.36	3
*Culicoides guttifer* Meijere	*Meijerehelea*	Sp19	500	0.38	576	0.25	10
*Culicoides histrio* Johannsen	*Meijerehelea*	Sp20	8	0.01	8	<0.001	25
*Culicoides homotomus* Kieffer	*Monoculicoides*	Sp21	1235	0.95	1582	0.70	6
*Culicoides pampangensis* Delfinado	*Ornatus* group	Sp22	44	0.03	44	0.02	21
*Culicoides oxystoma* Kieffer	*Remmia*	Sp23	43,570	33.34	55,579	24.61	2
*Culicoides shortti* Smith and Swaminath	*Shortti* group	Sp24	93	0.07	103	0.05	18
*Culicoides albibasis* Wirth and Hubert	*Trithecoides*	Sp25	2	<0.001	2	<0.001	30
*Culicoides anophelis* Edwards	*Trithecoides*	Sp26	460	0.35	803	0.36	9
*Culicoides flavescens* Macfie	*Trithecoides*	Sp27	595	0.46	916	0.41	8
*Culicoides gewertzi* Causey	*Trithecoides*	Sp28	9	0.01	31	0.01	22
*Culicoides parahumeralis* Wirth and Hubert	*Trithecoides*	Sp29	64	0.05	92	0.04	19
*Culicoides palpifer* Das Gupta and Ghosh	*Trithecoides*	Sp30	160	0.12	239	0.11	13
*Culicoides rugulithecus* Wirth and Hubert	*Trithecoides*	Sp31	6	<0.001	8	<0.001	26
Unidentified *Culicoides* spp.			2040	1.56	2863	1.27	
TOTAL			130,670	100	224,153	100	

**Table 3 insects-15-00701-t003:** Permutational multivariate analysis of variance (PERMANOVA) result showing effects of dates (DSIN, DCOS), environmental factors, and types of animals on community composition of using Bray–Curtis dissimilarity index as distance.

	Df	Sum of Squares	*R* ^2^	*F*	Pr (>F)	
DSIN	1	0.590	0.019	2.431	0.024	*
DCOS	1	0.564	0.018	2.324	0.027	*
Temperature	1	0.162	0.005	0.669	0.692	
Rainfall	1	0.656	0.022	2.702	0.016	*
Humidity	1	1.159	0.038	4.773	0.000	***
Wind speed	1	0.433	0.014	1.785	0.073	.
Cattle	1	1.390	0.046	5.721	0.000	***
Pigs	1	0.269	0.009	1.107	0.328	
Avians	1	0.720	0.024	2.966	0.006	**
Residual	98	23.809	0.800			
Total	107	29.757	1.000			

Significant codes: 0 ‘***’ 0.001 ‘**’ 0.01 ‘*’ 0.05 ‘.’ 0.1 ‘ ’ 1.

**Table 4 insects-15-00701-t004:** Summary of the effects of environmental factors on trap captures estimated by a generalized linear mixed model with trap as a random effect. Results for the total trap captures and each of the four major *Culicoides* biting midge species are presented. Collections were made using nine UV light traps set once a month from June 2020 to May 2021 in Tha Maka, Kanchanaburi, Thailand.

*Culicoides* sp.	Total		*C. imicola*		*C. peregrinus*		*C. arakawae*		*C. oxystoma*	
Environmental Factors	Coef.	*p*-Value	Coef.	*p*-Value	Coef.	*p*-Value	Coef.	*p*-Value	Coef.	*p*-Value
Temperature	0.257	<0.001	0.146	<0.001	0.184	<0.001	0.273	<0.001	0.403	<0.001
Rainfall	0.229	<0.001	−0.374	<0.001	0.429	<0.001	0.169	<0.001	−0.173	<0.001
Humidity	0.347	<0.001	0.487	<0.001	0.213	<0.001	0.348	<0.001	0.495	<0.001
Wind speed	−0.256	<0.001	−0.241	<0.001	−0.473	<0.001	−0.572	<0.001	−0.052	<0.001

**Table 5 insects-15-00701-t005:** Summary of the effects of animals on trap captures estimated by a generalized linear mixed model with collection date as a random effect. Results for the total trap captures and each of the four major *Culicoides* biting midge species are presented. Collections were made using nine UV light traps set once a month from June 2020 to May 2021 in Tha Maka, Kanchanaburi, Thailand.

*Culicoides* sp.	Total		*C. imicola*		*C. peregrinus*		*C. arakawae*		*C. oxystoma*	
Animal	Coef.	*p*-Value	Coef.	*p*-Value	Coef.	*p*-Value	Coef.	*p*-Value	Coef.	*p*-Value
Cattle	0.176	<0.001	0.447	<0.001	0.022	<0.001	−0.001	0.727	0.363	<0.001
Pigs	−0.100	<0.001	−0.274	<0.001	−0.412	<0.001	−0.208	<0.001	0.096	<0.001
Avians	−0.392	<0.001	−1.047	<0.001	−1.800	<0.001	0.021	<0.001	0.040	<0.001

## Data Availability

The data presented in this study are available on request from the corresponding author.

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
