# Peer review of "Seasonal Abundance and Diversity of Culicoides Biting Midges in Livestock Sheds in Kanchanaburi Province, Thailand"

_insects, 2024, doi:10.3390/insects15090701_

Round 1

Reviewer 1 Report

Comments and Suggestions for Authors

This paper looked at the seasonality and composition of Culicoides species in Thailand. This was a very large sampling effort that yielded a massive amount of population level data for biting midges in this area. The traps were also set at certain farms allowing for some investigation into potential host preferences. Environmental data was used to correlate the abundance of the most common Culicoides species to the factors such as humidity. While I can understand wanting to pull as much information from the substantial work done by the authors, I feel that some analyses in this paper overreached and were likely improperly analyzed. This is not to say that nothing here is worth publishing, however, I feel that the manuscript needs significant changes to correct some errors and refocus on the data that is important (i.e., tables 2 and 4; figures 3 and 4a).

Theoretically, a GLM should be able to pull variable importance from the type of livestock on each farm, however, there are far too much collinearity in this dataset. Each farm had a wildly different numbers and type of animal, but they seem to be analyzed as equivalent independent variables. For example, trap 8 had 30 cows and 2 doves (birds), yet these variables are treated as equivalent. If I have a huge pile of garbage next to vase of flowers, this might lead me to conclude that flowers attract large numbers of filth flies when the only thing driving this data is the trash. Trap 8 might be the most egregious example, but it illustrates the danger on analyzing these types of data without a more sophisticated model. Also, only one farm had pigs compared to the other livestock with far more replicates. As overall Culicoides abundance is the dependent variable tested, the animal density per farm, composition of the livestock, and distance from another farm (especially with a different livestock composition) will all matter. I just feel that overall, these data may not be worth analyzing the way the authors did. They very well can still show the abundance of the top four Culicoides species collected by trap and then overlay the livestock composition. This would allow the readers to see for themselves what is going on at each farm without adding false statistically significance to certain livestock species.

There is also a major issue with the NMDS analysis. It was not interpreted correctly, and the stress value indicates that the interpretability of the results presented are unlikely to be better than random chance.

Some acronyms are used without explanation.

Some of the figure and table legends need to be rewritten to increase readability.

Line items:

L31: swine

L35-37: change to something like “and other species accounted for the remaining 9,777 individuals (4.4%) for grammar.

L39: Culicoides midge catches

L42-43: number of

L47: add a colon after Diptera

L63: pathogens they transmit

Table 1: it is a bit hard to tell what livestock are on which farm. Would the authors consider adding lines between the rows to make this distinction easier to see?

L103: Based primarily on wing pattern, correct? Though maybe a small change, I think this distinction is important as it will help others who attempt the same thing.

L128: they could not be identified due to damage, or are they an unknown species?

Figure 2: please add how these photos were taken in the methods.

Figure 3: Presenting the data by animals is misleading here as almost every farm has multiple animal species. For example, the “bird” data looks more similar to the cattle data than to other avian data because birds were only identified on farms with cattle. Those two doves at that farm have nothing to do with the Culicoides species composition; the 30 cows are driving this as is shown in table 3.

Figure 3: the red and green used here will be hard for color blind people to see. The fact that the authors kept the key the same order as the data is very helpful though.

Figure 3: All is the mean abundance across all data, but it is unclear what livestock and poultry include. Please add this information to the figure legend if the figure is going to stay as it is.

In my opinion, a better way to present the data in figure 3 would be to show the abundance by trap/farm. The authors could then show small silhouettes of the animal on each farm underneath, and this would allow us to see

L143: again, the bird data is tied to the cattle data. I think the authors should just drop the “birds” from this study. The GLMM data does not support the C. peregrinus association with “birds”.

Table 5: Something is wrong with this table. The data here does not match the data in the paper. There were 108 total trapping events, not 100. Some of the rows don’t add up properly. What is wrong with “Farms with pigs”? This does not match any of the data reported in figure 2. Please ensure that this information is corrected.

Figure 4. This figure is unreadable. Pick a single diversity measure or average use the mean, not the per farm data. Also, what this shows is that relative abundance its inversely proportional to diversity measures which well known. I’m not sure how much this adds to the paper.

L196: The PERMANOVA supports that livestock, especially cattle, are driving Culicoides species composition in most cases and further supports my position on excluding that livestock GLMM.

L198-199: the axes of an NMDS ordination are entirely arbitrary and have no directionally. These can not be positive or negative in the way presented by the authors.

Figure 5: the 4 most abundance species should be listed in this figure by name not spID. It is very hard to pull meaning full information from this.

Figure 5: a stress value of 0.25 lies somewhere between “Could be dangerous to interpret” and “Samples placed essentially at random”; see Clarke, K.R. 1993. Non-parametric multivariate analyses of changes in community structure. Australian Journal of Ecology 18:117-143. Specifically, pg. 126. Any conclusions about this data are likely not true.

L203: If you start a sentence with a species name, spell it out.

L217-223: is this type of seasonality common in similar climates?

L233: I think its higher, but not much, maybe like 2.5m/s. It also depends on the species.

L237-238: this seems like the wrong citation for this information.

L246-247: I do not know what the authors are trying to say here.

L252: do the authors mean the number of midges collected when they say trap size?

L262: Culicoides

L264: this is not the way you abbreviate a subgenus. Just mention that these species are in the subgenus Avaritia.

L265-269: A Culicoides female with travel up to 2km while host seeking. Are the fields that far away?

L270-272: consider revising for clarity.

L280-294: I’m not sure that the number of times a species is found in a trap should be used to make all of these conclusions, especially as most farms had a wide variety of animal species/were relatively close to another trap. At the very least, the data are not as clean as what is presented here. For example, C. oxystoma was collected regularly at farms with chickens, for more than pigs. Does this mean that this species prefers chickens over pigs? The assessment that C. peregrinus likely prefers cattle, while C. arakawae appears to be attracted to birds (which is actually not supported by the GLM in table 3) could be true. However, the much more interesting finding here that is not discussed was why was C. peregrinus not collected from the pig farm.

L296-267: why did cattle have the greatest effect on species diversity.

L304-310: Not every abundance species should be a suspected vector if there is no evidence supporting the vector host interaction. The authors then go on to say that while abundant around pig, there is no evidence that C. oxystoma feeds on them. This paragraph is just very messy.

L313: Capitalization.

Comments on the Quality of English Language

There are grammatical issues, primarily in the discussion. I did not mention every single one in the line items, but I would recommend a thorough review.

Author Response

To: Reviewer 1

Thank you for your comments. We made a major revision according to your comments. Please find an attached file. We try to explain how we responded to comments and suggestions by the two reviewers. To enhance readability, we also attached a revised manuscript, in which red font color indicates our major changes.

Reviewer 2 Report

Comments and Suggestions for Authors

Line

Please review the following points:

23 – Add the author's name.

25 – Change "types" to "animals."

41 – Provide reasons for "Sheds containing cattle produced more Culicoides."

48 – Add information about medico-veterinary diseases.

79 - Specify how air temperature and humidity were recorded; clarify whether microclimatic devices were used.

84 - Specify the instruments used.

89 - Include UV light traps.

Table 1. Include the scientific names of the animals and birds.

96 – Provide information about adult Culicoides.

98 – Explain how subsampling was done and add a reference.

101 – Put "Culicoides" in italics.

102 – Mention the concentrations of ethanol.

128 - Clarify why the individuals were not identified; was it due to a lack of keys or descriptions?

131 – Replace "captures" with "catch data."

134 – Clarify about female wings.

136 – Explain the male catch data.

139 – Rewrite the legend as "Relative abundance of Culicoides spp in the presence of animals."

147 - Clarify whether the number of cattle or the body size affects the catch data.

157 – Delete "the greatest number."

Table 3 – Replace "summary of the effects" with "influence."

Table 4 – Redraft the legend.

Table 5 – Remove "farm" from all the columns.

178 & 179 – Redraft the sentence.

184 – Explain what is meant by "opposite pattern."

Graph (a) – Cluster the nine traps based on the types of animals.

Graph (b) to (e) – Delete these graphs.

256 – Explain what is meant by "inhibitory level."

262 – Correct the spelling of "Culicoides".

322 – Put "Culicoides peregrinus" in italics.

Comments on the Quality of English Language

Language needs improvement. 

Author Response

To: Reviewer 2

Thank you for your comments. We made a major revision according to comments by the reviewer 1. Please find an attached file. We try to explain how we responded to comments and suggestions by the two reviewers. To enhance readability, we also attached a revised manuscript, in which red font color indicates our major changes. 
